# Evolving Clinical Utility of Liquid Biopsy in Gastrointestinal Cancers

**DOI:** 10.3390/cancers11081164

**Published:** 2019-08-13

**Authors:** Richard A. Jacobson, Emily Munding, Dana M. Hayden, Mia Levy, Timothy M. Kuzel, Sam G. Pappas, Ashiq Masood

**Affiliations:** 1Department of Surgery, Rush University Medical Center, Chicago, IL 60612, USA; 2Division of Hematology/Oncology and Cell Therapy, Rush University Medical Center, Chicago, IL 60612, USA; 3Rush Precision Oncology Program, Rush University Medical Center, Chicago, IL 60612, USA

**Keywords:** gastrointestinal cancer, liquid biopsy, circulating tumor cell, circulating tumor DNA

## Abstract

Room for improvement exists regarding recommendations for screening, staging, therapy selection, and frequency of surveillance of gastrointestinal cancers. Screening is costly and invasive, improved staging demands increased sensitivity and specificity to better guide therapy selection. Surveillance requires increased sensitivity for earlier detection and precise management of recurrences. Peripherally collected blood-based liquid biopsies enrich and analyze circulating tumor cells and/or somatic genomic material, including circulating tumor DNA along with various subclasses of RNA. Such assays have the potential to impact clinical practice at multiple stages of management in gastrointestinal cancers. This review summarizes current basic and clinical evidence for the utilization of liquid biopsy in cancers of the esophagus, pancreas, stomach, colon, and rectum. Technical aspects of various liquid biopsy methodologies and targets are reviewed and evidence supporting current commercially available assays is examined. Finally, current clinical applicability, potential future uses, and pitfalls of applying liquid biopsy to the screening, staging and therapeutic management of these diseases are discussed.

## 1. Introduction to Liquid Biopsy

Effective management of gastrointestinal tract cancers depends on early detection, accurate staging, therapy selection, and timely surveillance. All components of management hinge on the presence or absence of systemic disease, which was historically detectable only after the development of bulky metastases. Such metastases are currently detected with a combination of serum markers, radiography, and invasive procedures such as endoscopy.

Solid tumors have long been known to shed malignant cells and genomic material into the circulation antecedent to the development of macrometastases [1]. In recent decades, advances in laboratory techniques have permitted the detection of microscopic tumor material in remote body fluids, termed liquid biopsy [2]. It is the goal of this review to summarize and clarify current evidence informing the use of liquid biopsy in esophageal, gastric, pancreatic, and colorectal cancer. Data suggestive of potential clinical benefits of liquid biopsy are summarized based on the stage of cancer management impacted (Figure 1).

## 2. Technical Concerns in Liquid Biopsy

Until recently, liquid biopsy of peripheral blood lacked the sensitivity and specificity required for clinical utility in cancer management [3]. Advances in both detection technology and understanding of the biological underpinnings of cancer dissemination have made liquid biopsy a viable clinical tool. Novel microfluidic and miniaturization technologies have enabled high throughput evaluation of clinical samples [4]. The most common analytical targets of liquid biopsy are circulating tumor cells (CTCs) and circulating tumor DNA (ctDNA). Other novel targets include exosomal DNA [5], microRNA [6], and tumor-educated platelets [7]. Broadly, liquid biopsy protocols involve the collection of peripheral blood, followed by enrichment, then specific detection of the analyte of choice, as illustrated in Figure 2.

CTC-based assays exclude cells grossly unlike CTCs based on size and surface markers. This enrichment step is required due to the relatively miniscule amount of circulating tumor material present in a standard 2–4 mL collection of peripheral blood. Following enrichment, conclusive identification of CTCs requires either cytologic or genetic analysis. Cytologic detection with surface markers allows for the further assessment of cell viability and, in some cases, improved prognostication compared to genetic detection [8]. Notably, only a small percentage of CTCs detected in peripheral blood have the potential to seed metastatic disease; most die within hours [9]. The ideal detection system would maximize sensitivity by probing for multiple markers, quantify viable CTCs, and have the ability to characterize epithelial-to-mesenchymal transition (EMT) status via nucleic acid amplification.

Genetic material is released into circulation through cell death or active processes such as exosome release. ctDNA-based liquid biopsy excludes benign cell-free DNA to identify tumor-derived ctDNA, probing circulating oligonucleotides that harbor tumor-specific mutations. ctDNA most commonly occurs as fragments on the order of 100–200 base pairs, which are shorter on average than fragments of benign circulating DNA. Assays performed in plasma offer superior sensitivity to serum-based assays due to increased release of benign cell-free DNA encountered during the processing of serum [10]. Such assays may detect tumor-specific point mutations, methylation patterns, or copy number changes, among other targets.

Intratumoral heterogeneity is a primary reason for treatment failure owing to drug resistance and therapeutic failure [11]. Therefore, management of systemic disease based solely on a biopsy of solid tumor tissue is vulnerable to sampling error which, if not taken into consideration, can lead to inappropriate therapeutic interventions. Liquid biopsy may reflect current and complete genomic architecture of invasive subclones with the potential to seed metastatic disease; therefore, acquired resistance following therapeutic interventions may offer a broader understanding of tumor biology compared to tumor tissue genomics [12]. In the coming years, we anticipate available laboratory technology will continue to improve, as will the dynamics of all stages of cancer management based on liquid biopsy.

## 3. Esophageal Cancer (EC)

CTCs and tumor microemboli (clusters of tumor-derived cells) have been detected in the peripheral blood of patients with both subtypes (adenocarcinoma and squamous cell carcinoma) of EC [13,14]. CTCs in EC patients are found at a relative concentration of 1 per 5–10 million white blood cells [15]. As such, sample enrichment based on either immunolabeling of surface antigens or physical characteristics is required. Following enrichment, CTCs are identified cytologically or via RT-PCR for epithelial cellular adhesion molecule (EpCAM), carcinoembryonic antigen (CEA), or cytokeratin [16].

Current ctDNA-based liquid biopsy generally targets circulating oligonucleotides containing point mutations or alterations in methylation. Digital PCR-based detection methods offer high sensitivity for ctDNA containing mutations already identified in the primary tumor; however, assays that instead employ next-generation sequencing cast a broader net and are more likely to detect mutations discordant from the primary tumor genotype.

### 3.1. Screening

Esophageal cancer screening is invasive and limited to high-risk patients with persistent symptoms of acid reflux [17]. Unfortunately, capture of the relevant population and thus utilization of screening is low. A study of over 900 patients demonstrated that 93% of patients diagnosed with EC had no prior diagnosis of premalignant Barrett’s metaplasia, nor any screening [18]. Another large dataset demonstrated that 40% of patients presented with incurable late-stage disease at the time of diagnosis [19]. Despite this, no screening test is currently employed for EC in average-risk patients [20].

Liquid biopsy may represent a sensitive and minimally invasive alternative to endoscopic screening. A small study of CTC-based liquid biopsy differentiated squamous cell EC patients from healthy controls with 86.3% sensitivity and 90.3% specificity [21]. ctDNA is less sensitive, as it is detectable in only 58% of patients without metastatic disease, however, sensitivity increases with late disease stage [22]. High sensitivity and decreased invasiveness make liquid biopsy an attractive strategy, however, large prospective trials are required to validate liquid biopsy as a means of screening for EC.

### 3.2. Staging

Current staging is radiographic, involving PET-CT and endoscopic ultrasound. Nearly 50% of patients without evidence of disseminated disease at the time of curative-intent therapy develop distant metastases within five years, mainly due to hematogenous dissemination [23]. Thus, as would be expected, existing staging modalities are insensitive for micrometastatic disease at the time of curative-intent therapy.

Clearly, room for improvement exists in staging, and liquid biopsy has the potential to improve on current modalities. Prospective studies of 74–106 patients with EC demonstrated that the presence of CTCs prior to treatment correlated positively with the degree of local invasion, presence of nodal disease [24], and presence of distant metastases and correlated negatively with two-year progression-free survival [10] and overall survival [25]. In addition, later-stage patients express a higher proportion of mesenchymal-type CTCs than early-stage patients [26]. One study demonstrated that CTCs may be present even in patients with pathologic N0 disease, implying that liquid biopsy could be used to detect disseminated disease in patients with negative nodes [27].

ctDNA-based liquid biopsy has the potential to improve prognostication in both squamous and adenocarcinoma. While the total level of cell-free DNA (cfDNA) is not correlated with overall survival [28], ctDNA assays can predict poor prognosis through the detection of specific mutations or methylation patterns. Multiple small studies have detected ctDNA molecular signatures that can stratify EC patients into survival clusters based on implied tumor biology [29]. In patients with squamous cell cancer, the peripherally detected MSH2 hypermethylation promoter was associated with inferior disease-free survival [30].

The above studies suggest that liquid biopsy can further stratify patients with clinically apparent localized EC based on endoscopy and radiography. Ideally, clinical trials to refine indications for neoadjuvant and adjuvant therapy could be enriched with “high-risk” individuals rather than all-comers with liquid biopsy.

### 3.3. Response to Therapy/Surveillance

Management is stage-dependent, but in a curative setting involves a multimodality approach with chemo/radiation therapy and surgical resection of the primary tumor. One study demonstrates that persistent CTC positivity following radiation therapy may be a negative prognostic factor [17]. Absolute quantity of CTCs has been demonstrated to decline following surgical resection and the extent of the decline is correlated with the development of metastases [28]. Data from a small study (n = 44) suggests that CTCs may complement traditional serum markers such as CEA in surveillance settings [31]. In patients treated with chemoradiation alone, pre-radiotherapy CTC positivity was associated with a 2.5 (0.9–7.2) relative risk of recurrence within two years. Post-radiotherapy CTC positivity had a relative risk of 3.7 (1.4–9.8) for recurrence [24].

Multiple studies revealed that somatic mutations detected in ctDNA were discordant with mutations in the tumor tissue in 13–39% of patients [32]. The Personalized Antibodies for GastroEsophageal Adenocarcinoma (PANGEA) trial highlighted the significance of tumor heterogeneity and the potential of ctDNA-based liquid biopsy to guide precision therapy [33]. Targeted therapy in such patients may confer a survival benefit that would not be available if the genomic analysis were limited to the tumor tissue. Another group demonstrated that acquired *MET* amplifications detected in ctDNA can serve as markers of resistance to trastuzumab and novel HER-kinase inhibitors in adenocarcinoma [34].

Response to therapy, determination of resistance to targeted treatment, and frequency of surveillance may be facilitated by liquid biopsy as well. Further prospective studies in which patients receive personalized therapy based on specific mutations identified in ctDNA are necessary to confirm the utility of ctDNA in clinical practice. Whether patients should undergo serial liquid biopsies throughout the course of treatment to provide clinically relevant trends are another area that needs to be explored in a prospective fashion.

## 4. Gastric Cancer (GC)

### 4.1. Techniques

Both CTCs and ctDNA have been detected in the peripheral blood of patients with gastric cancer. One study showed that >2 CTCs per mL of peripheral blood had 85.3% sensitivity and 90.3% specificity for distinguishing 116 patients with known gastric cancer from 31 healthy controls. The false positive rate in this study was 6.5% (2/31) [35]. The concentration of CTCs in patients with known GC is 1 cell per 10^7^ white blood cells collected [36]. CTCs have been detected at the epithelial, intermediate, and mesenchymal phases of EMT [37]. In addition, a meta-analysis including nearly 1200 patients demonstrated that ctDNA was detectable in the peripheral blood of patients with gastric cancer [38].

### 4.2. Screening

No average-risk screening for GC is currently employed in the United States. In Japan, where the incidence is higher, nationwide endoscopic screening has diminished GC-related mortality over the past decade [39]. Based on a review of the current body of literature, stronger evidence exists for ctDNA and miRNA as screening biomarkers for GC than CTCs [40]. However, a small trial demonstrated that sensitivity for early (stage I) disease was below 50% [41]. Greater sensitivity would be required for CTCs to be a useful screen for early-stage disease.

No current tests offer the sensitivity and specificity that would be required to supplant endoscopic screening in high incidence locations such as Japan, or to justify broad screening in the United States. The CancerSEEK platform based on ctDNA demonstrated nearly 70% sensitivity to all-comers, however, sensitivity for stage I disease was extremely low with currently available assays [41]. More studies are needed to confirm whether ctDNA- or miRNA-based assays could replace endoscopic screening or whether employing both modalities in combination could improve early detection and outcomes.

### 4.3. Staging

Detection of CTCs and ctDNA is a promising adjunct to current established staging modalities in gastric cancer. A study of 44 patients demonstrated that the CTC count at diagnosis is predictive of advanced stage, peritoneal dissemination, presence of metastases, and poor survival [42]. Furthermore, a recent meta-analysis of 16 studies demonstrated that quantitative ctDNA level correlated with the above outcomes [38]. Another benefit of ctDNA in staging is that epigenetic markers, including CpG methylation, appear to be clinically relevant predictors of TNM stage [38].

A well-established treatment for advanced esophagogastric cancer is anti-HER2 antibodies in tumors with HER2 amplification or overexpression. However, resistance to these antibodies poses a barrier to effective therapy. A prospective study of 88 patients demonstrated discordance in *HER2* amplification status between primary tumors and CTCs in patients with metastatic GC [43]. As targeted therapy with trastuzumab is designed to treat systemic disease, this may indicate that tissue genomics and liquid biopsy have complimentary roles in detecting actionable alterations. Therefore, obtaining both tissue and liquid biopsies may be a reasonable approach in the clinical setting.

### 4.4. Response to Therapy/Surveillance

Although not broadly employed yet in clinical environments, it appears that the most actionable current use for liquid biopsy is likely in monitoring response to therapy and acquired resistance and guiding targeted therapy with the ability to detect sensitivity and resistance patterns. A small study demonstrated discordant genotypes acquired from tissue biopsy between primary and metastatic lesions, leading to clinically significant treatment reassignment [44]. That same study demonstrated that metastatic lesions and ctDNA were concordant in 87.5% of cases, which suggests that ctDNA could substitute for tissue biopsy of metastatic lesions to guide systemic therapy. In a similar fashion to biopsy data from the primary tumor, peripheral levels of BRCA1 and thymidylate synthetase mRNA predicted sensitivity to docetaxel (positive association) and pemetrexed (negative association) in a study of 150 patients [45]. Trastuzumab resistance in metastatic HER2-positive disease can also be monitored with serial liquid biopsies and therapeutic regimens can be adjusted accordingly [46]. A similar example of this is illustrated in a study that trialed a pan-HER kinase inhibitor (afatinib) and found that resistance to this drug was correlated with tumor cells that were lacking EGFR amplification or with MET amplification [47].

Another GC therapy currently under clinical investigation is anti-EGFR agents in patients with EGFR amplification. While determining the utility of these drugs, it was discovered that ctDNA could be utilized to reveal pre-existing or acquired genomic events that ultimately lead to anti-EGFR resistance [48].

A final role for liquid biopsy in predicting overall response to therapy in GC is one demonstrated throughout most surgically resectable gastrointestinal cancers. This is the correlation between elevations of CTC levels in post-treatment patients with overall poor prognosis. This, combined with other prognostic data, provides an even more accurate prognosis [49].

## 5. Pancreatic Adenocarcinoma (PDAC)

### 5.1. Techniques

Promising research has demonstrated a potential role for liquid biopsy in the diagnosis and management of PDAC. In patients with PDAC, both CTCs and ctDNA have been identified in peripheral blood. Enrichment is challenging in PDAC due to small numbers of CTC in peripheral blood and pilot studies demonstrated that 7.5 mL whole blood is required for adequate sensitivity; in addition, cell size-based isolation (ISET) was more sensitive than the Cellsearch system [50]. Following enrichment, CTCs are analyzed using immunocytochemistry, FISH, and whole genome sequencing.

A study of 34 patients demonstrated that PDAC patients released ctDNA into the circulation at a rate that was orders of magnitude higher than healthy controls and genomic and epigenetic mutations were detectable in 5 mL of whole blood [22]. The optimal technique for enrichment and analysis of both CTCs and ctDNA in PDAC is a topic of ongoing laboratory investigation.

### 5.2. Screening

Pancreatic cancer is currently the 4th leading cause of cancer death in the United States, primarily due to late presentation of illness, demonstrating an unmet need for effective PDAC screening. A recent study suggested that PDAC takes at least 15 years to develop from initial mutations to metastatic cancer [51], highlighting the potential for utilizing screening methods for early diagnostics if the sensitivity of currently available assays can be improved.

CTCs are present in patients across all stages of PDAC, more so in patients with advanced disease. However, they are limited for screening use at present due to high rates of false positives in non-malignant disease states, such as chronic pancreatitis (up to 63%) [52]. One study of 68 patients demonstrated only 68% specificity in distinguishing PDAC (particularly PDAC) from other pancreatic conditions and healthy patients [53].

Four principal mutations have been identified in ctDNA as most commonly found in PDAC: *KRAS, CDKN2A, SMAD4*, and *TP53*. A pilot study of exome sequencing demonstrated *KRAS* as the most commonly mutated gene in ctDNA (90% to 95% of cases) [54]. As with CTCs, specificity for PDAC is low in ctDNA screening studies. One study identified circulating DNA with *KRAS* mutations in up to 20% of chronic pancreatitis patients [16]. However, ctDNA-based liquid biopsy may still have a significant role in PDAC screening. A study combining the detection of ctDNA *KRAS* mutations with CA 19-9 levels differentiated PDAC (47 patients) from chronic pancreatitis (31 patients) with a sensitivity of up to 98% and a specificity of 77% [55]. While peripheral biomarkers collected in liquid biopsy such as CTCs and ctDNA in PDAC are not yet specific or sensitive enough to act as a screening tool, technical advances and computer-guided algorithmic predictions used in combination with traditional biomarkers are promising approaches that should be explored further.

### 5.3. Staging

Current staging for PC is done radiographically, with patients’ resectability status determined by presence of local invasion, involvement of major peripancreatic vessels, and the presence of remote disease. The most evident potential benefit of liquid biopsy lies in prognostic indicators with the potential to guide management and determine resectability. A meta-analysis of 16 studies regarding the use of CTCs and disseminated tumor cells (CTCs in bone marrow) as prognostic indicators demonstrated that detection of CTCs in peripheral blood and CTCs in bone marrow is associated with both progression-free and overall survival [56].

As for the role of ctDNA, a meta-analysis of 18 studies including 1243 patients determined that the presence of ctDNA with *KRAS* or ERBB2 mutations in peripheral blood significantly impacted overall and progression-free survival. Furthermore, specific mutations, hypermethylation, and a high absolute quantity of ctDNA were associated with inferior survival [57]. Potential improvements in staging lie in somatic alterations identified in ctDNA (*KRAS*) in the peripheral blood or portal blood. One study demonstrated that a *KRAS* mutation in the portal venous circulation was associated with increased risk of developing metastases [58]. Furthermore, in a subset of patients with resectable disease who underwent pancreaticoduodenectomy, peripherally detected *KRAS* mutations were present in 31%. *KRAS* mutation-positive patients demonstrated an overall survival of 13.6 months compared to 27.6 months in those with wild-type cancers [59]. Interestingly, peripherally detected *KRAS* mutations were a worse prognostic indicator when patients with all stages of disease were analyzed [60].

### 5.4. Response to Therapy/Surveillance

Even after curative intent resection, five-year survival in PDAC is low. Contributing to this is recurrence, with a majority of resected patients experiencing recurrence within two years of surgery. Frequent physical exams, CT imaging, and CA 19-9 levels are currently utilized, however, no clear guidelines for surveillance are currently in place.

Liquid biopsy may predict overall survival and response to therapy in PDAC. An example of this is CTC positivity, which dropped from 80.5% of patients to 29.3% after a cycle of fluorouracil-based chemotherapy in 41 patients with advanced PDAC [61]. Berger et al. demonstrated that in 20 therapy naïve patients with ctDNA-positive (*KRAS* or *TP53* mutations) metastatic PDAC who underwent chemotherapy, detectable mutations significantly decreased during treatment and the resulting levels correlated with progression-free survival [62]. The same study provided proof-of-principle that genomic profiling of mutations detected in peripheral ctDNA could guide targeted therapy.

Further research is warranted to demonstrate the utility of liquid biopsy in monitoring PDAC recurrence. One study demonstrated that specific CTC phenotypes (high expression of CD44, CD133, and aldehyde dehydrogenase) are predictive of recurrence [63]. Higher post-treatment levels of overall plasma ctDNA were found to correlate with both recurrence and metastases [64]. Furthermore, one study showed that ctDNA detected recurrences 6.5 months earlier than CT imaging among 77 patients [65]. Larger prospective studies are needed to confirm these assays before employing them routinely in clinical practice.

## 6. Colorectal Cancer

### 6.1. Techniques

The role of liquid biopsy in colorectal cancer (CRC) is evolving rapidly. CTCs, ctDNA, and exosomal DNA have been isolated from the blood of patients with CRC. Location of the blood-draw impacts the sensitivity of liquid biopsy in CRC, as blood from the portal system is enriched with tumor material compared to peripheral venous blood [66]. A small study suggested that ctDNA-based liquid biopsy was more sensitive for the detection of metastatic colorectal cancer than CTCs, however sensitivity is dependent on specific protocols [67]. The same study demonstrated detectable ctDNA in 20/20 patients, with discordance to the primary tumor genotype in 15%.

Detection limits reach as low as 3 CTCs per 2 mL whole blood [68]. CTCs in CRC patients frequently have a genotype distinct from tissue tumor cells, as demonstrated by the up- or down-regulation of hundreds of genes involved in metabolism, epithelial-to-mesenchymal transition, and cell survival [69]. In a paradigm that could potentially impact treatment decisions in the future, CTCs collected serially from a single patient were isolated, grown as cell lines, and genetically analyzed. Chemoresistance patterns were sequentially determined ex vivo on these lines and correlated with in vivo tumor biology.

Total cell-free DNA and ctDNA were elevated in the peripheral blood of CRC patients compared to both healthy controls and patients with inflammatory bowel disease [70]. The half-life of ctDNA in CRC is roughly 2.5 h, allowing for a relatively rapid determination of response to therapy [71]. ctDNA can be detected using either targeted [22] or untargeted [72] approaches depending on the indication for the assay. Both variant alleles and epigenetic markers (CpG island methylation) are relevant in the detection and interpretation of ctDNA in CRC [73].

### 6.2. Screening

CRC is unique among gastrointestinal malignancies in that a relatively mature screening program exists for average-risk patients. Colonoscopic screening is effective yet invasive, associated with poor compliance from patients, and is labor- and resource-intensive. Less invasive means of screening, such as fecal immunochemical tests and DNA tests (e.g., Cologuard), have been developed and deployed clinically in recent years. It is important to note that any positive noninvasive screen for colon cancer is followed up with a diagnostic colonoscopy. If sufficient sensitivity and specificity can be achieved, liquid biopsy would save disease-free patients the cost, inconvenience, and morbidity of screening colonoscopy. No liquid biopsy-based assays were recommended for CRC screening in the 2018 Guideline Updates from the American Cancer Society, however, as trial data are collected and as technology improves, such tests may have a greater role in CRC screening in the future.

Multianalyte detection systems (CancerSEEK) analyzed specific mutations in ctDNA along with other traditional peptide markers for CRC and other cancers in preclinical experiments. Data collected in this study were run through a supervised machine-learning algorithm that was able to identify CRC positivity in 100% of patients with known disease [41]. CTC-based liquid biopsies designed for screening depend on early shedding of identifiable adenomatous cells prior to the development of an invasive cancer. Early trial data suggest that commercially available FirstSight CTC-based assays have superior sensitivity (77%) for precancerous lesions compared to other minimally invasive means for CRC screening [74].

### 6.3. Staging

Indications for adjuvant chemotherapy are well-defined in patients with stage III disease, however, patients with stage II disease require further stratification to determine whether systemic therapy is indicated. The ability to better identify those patients that would benefit from systemic therapy is critical to improving survival and limiting toxicity and should be investigated further. A meta-analysis of 11 studies demonstrated that ctDNA positivity at the time of diagnosis is associated with inferior overall survival in colon cancer, however, the strength of this association varies with timing and technique [75]. Exosomal mircoRNA-21 positivity was correlated with poor disease-free and overall survival in stage II and III colon cancer [76]. In contrast to colon cancer, pre-treatment ctDNA positivity in rectal cancer did not correlate with overall survival. However, that same study demonstrated that post-resection positivity was extremely predictive of recurrence and could be used to identify stage II patients that would benefit from chemotherapy [77].

Current staging and surveillance systems poorly identify patients who may benefit from invasive regional, and even distant, curative therapies, such as cytoreductive surgery (CRS) and hyperthermic intraperitoneal chemotherapy (HIPEC). Indications for such therapy are currently based on imaging findings in patients who are already symptomatic. A pilot study of 14 patients suggested that postoperative CTC-positive patients with suspicion of peritoneal disease would benefit from CRS/HIPEC in terms of overall survival [78]. The indications for such regional therapy are currently ill-defined and further investigation into liquid biopsy as an adjunctive staging measure is indicated.

The presence of mutations such as *KRAS* that impact the efficacy of targeted therapy plays a major role in the staging of CRC. Digital PCR specific for *KRAS* and *BRAF* mutations was able to detect variant alleles in the ctDNA of 66% of patients with locally advanced rectal cancer that harbored these mutations in surgically resected specimens. On the other hand, 15% of preoperative patients had *KRAS* mutations detected via ctDNA that were not detected during endoscopic biopsy [79]. Some experts currently recommend the use of targeted ctDNA-based liquid biopsy for *RAS* mutations when biopsy of the primary tumor is unfeasible [80].

### 6.4. Response to Therapy/Surveillance

Liquid biopsy has demonstrated promise in detecting minimal residual disease or recurrence during the surveillance of colon cancer. Standard post-treatment surveillance requires the use of non-specific serum markers (CEA), endoscopy, and an abdominopelvic CT scan. A well-designed study of 230 patients determined that ctDNA with either a *KRAS*, *TP53*, or *APC* mutation was sufficient evidence of residual disease after curative-intent therapy [81]. Supporting this, a recent prospective study of 125 stage I–III colon cancer patients reported that, following definitive treatment, patients with persistent ctDNA during surveillance had a 40 times higher risk of recurrence that those with negative ctDNA [82].

Most patients with CRC are treated with 5-fluorouracil-based chemotherapy. Targeted agents such as panitumumab and cetuximab are routinely added to these regimens if the tumor is *RAS* wild-type. However, most patients go on to develop resistance to these agents. Thus, treatment decisions in metastatic CRC hinge on the presence of treatment resistance. One well-designed study demonstrated that ctDNA-based liquid biopsy was extremely sensitive (>85%) for primary resistance to *HER2* inhibitors [83]. Presumably, the ease of access of liquid biopsy could lead to increased sensitivity and earlier detection of treatment resistance compared to radiographic surveillance alone. This idea was supported by observations that demonstrated the superiority of liquid biopsy to tissue biopsy in determining resistance patterns, owing to tumor heterogeneity [84]. In fact, a randomized controlled trial of 254 patients used ctDNA to identify a subgroup of patients with anti-EGFR resistance that benefited from alternative targeted therapy [85]. Response to systemic therapy could also be monitored with CTC-based liquid biopsy and persistent positivity following adjuvant mFOLFOX in stage III colon cancer patients was a poor prognostic factor [86]. Notably, CTCs may evade detection through treatment-mediated selection over time. CTC’s, as detected by the CellSearch system, were observed to decline significantly in stage IV patients treated with bevacizumab who in fact had progressive disease [87]. Taken together, liquid biopsy offers the possibility to detect disseminated colon cancer earlier than current surveillance techniques with the added advantage of informing targeted therapy.

Advances in regional therapy for rectal cancer in particular have made distant recurrences the primary cause of mortality. A point of clinical equipoise is whether rectal cancer patients benefit from adjuvant systemic therapy after neoadjuvant chemoradiation and surgical resection. A well-designed cohort study with 159 patients demonstrated that the presence of ctDNA after resection is indicative of residual disease and a high probability of recurrence. In these patients, treatment with adjuvant systemic chemotherapy significantly improved OS [77]. Another study of 130 patients with stage I–III CRC suggested that serial ctDNA analysis with ultradeep sequencing could predict recurrence as early as 30 days after resection, which is much sooner than the capabilities of current radiographic, endoscopic, and CEA-based surveillance [82,88].

## 7. Summary

As a screening tool, liquid biopsy has the potential to save the cost and morbidity of invasive endoscopic screening procedures in diseases such as CRC and improve early detection of diseases such as pancreatic adenocarcinoma. It appears ctDNA-based liquid biopsy is generally more sensitive for the early detection of gastrointestinal cancers than CTC-based assays. While surgical techniques and outcomes continue to improve, procedures such as esophagectomy, gastrectomy, and pancreaticoduodenectomy still come with a high risk of morbidity and mortality. The ability to accurately stage patients with micrometastatic disease could save patients, who would not benefit from upfront resection, the morbidity inherent to such operations. Presence of CTCs may be an indicator of unresectability in certain gastrointestinal tumors, whereas the presence of ctDNA in determining resectability is less clear. Both CTC- and ctDNA-based liquid biopsy offer broad advantages over currently employed surveillance of gastrointestinal tumors using radiography, endoscopy, and tumor markers. Regarding the selection of systemic therapy regimens, ctDNA is generally more predictive of response to therapy and more useful in identifying actionable mutations for targeted therapy. Improvements in surveillance would benefit thousands of patients each year with locally advanced disease who may be candidates for systemic or extensive regional therapy, including cytoreductive surgery and hyperthermic intraperitoneal chemotherapy (HIPEC) [89].

A catalog of actively enrolling clinical trials investigating liquid biopsy in gastrointestinal cancers is shown in Table 1. In addition, partnerships between academia and industry have held an important role in the optimization of liquid biopsy assays. The goals of such partnerships are lofty, including ctDNA-based screening liquid biopsy for multiple gastrointestinal cancers in a single assay. One promising example is the Circulating Cell-free Genome Atlas study—a collaboration between GRAIL, Inc. and major academic institutions [90]. Validation of such technology would surely have far-reaching effects in access to cancer screening and outcomes.

Liquid biopsy technology has improved alongside basic understanding of cancer biology in recent years. The convergence of ideas from these two fields has led to the development of clinical-grade assays with the potential to impact all stages in the management of gastrointestinal cancers. Although still evolving, if applied correctly, existing and future technologies will hopefully improve access to care, management, and outcomes across all gastrointestinal cancers.

## Figures and Tables

**Figure 1 cancers-11-01164-f001:**
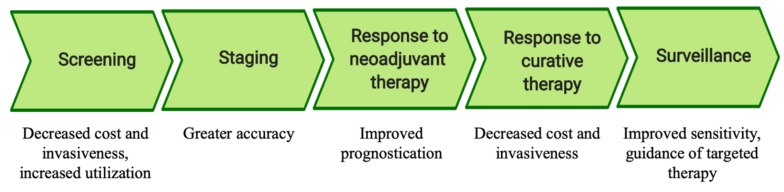
Potential applications of liquid biopsy by stage of cancer management.

**Figure 2 cancers-11-01164-f002:**
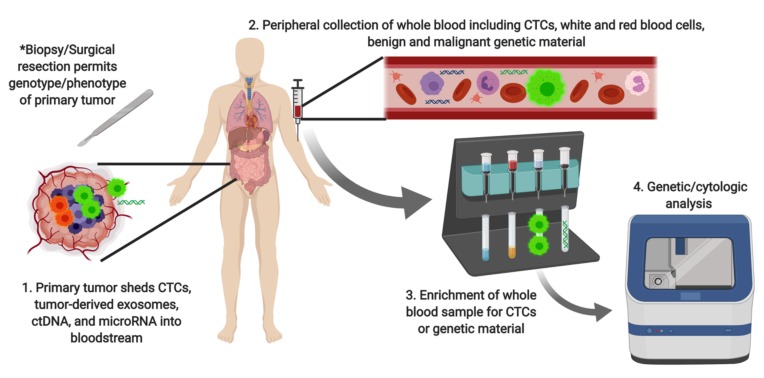
Schematic of liquid biopsy technique. CTC: circulating tumor cell; ctDNA: circulating tumor DNA.

**Table 1 cancers-11-01164-t001:** Selected actively enrolling clinical trials of liquid biopsy. LB: liquid biopsy; EUS: Endoscopic ultrasound.

Population	Phase	Hypothesis	Trial #
Resectable EACA	Staging	LB predicts response to neoadjuvant	NCT02812680
Resectable ESCC	Staging	LB predicts response to neoadjuvant	NCT03005314
Resectable GC	Response to therapy	CTC predict response to surgery and adjuvant	NCT03156777
Resectable GC	Response to therapy	ctDNA predict response to surgery and adjuvant	NCT02887612
High risk for PDAC	Screening	EUS-guided portal vein liquid biopsy	NCT03821909
High risk for PDAC	Screening	Detection of CTCs/ctDNA for PDAC	NCT02072616
Stage IV PDAC	Response to therapy	CTCs predict response to chemotherapy	NCT03033927
Unresectable CRC	Response to therapy	ctDNA level predicts progression	NCT03844620
Unresectable CRC	Response to therapy	LB detect acquired RAS mutation	NCT03401957
Resectable CRC	Response to therapy	CTC decrease following resection	NCT03256084
Resectable CRC	Response to therapy	CTC predict recurrence in liver metastasectomy	NCT03295591

CTC: circulating tumor cell; ctDNA: circulating tumor DNA; EACA: Esophageal adenocarcinoma; ESCC: Esophageal squamous cell carcinoma; GC: Gastric cancer; PDAC: Pancreatic ductal adenocarcinoma; CRC: Colorectal cancer.

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
