# Peer review of "Evolving Clinical Utility of Liquid Biopsy in Gastrointestinal Cancers"

_cancers, 2019, doi:10.3390/cancers11081164_

Round 1

Reviewer 1 Report

The authors have addressed my comments. Minor suggestions-

·        Lines 90-92: “CTCs are identified via nucleic acid amplification of mRNA for epithelial cellular adhesion molecule (EpCAM), carcinoembryonic antigen (CEA) or cytokeratin (16), or further immunolabeling of specific surface antigens”  -- consider reworking this sentence for accuracy

·        Line 309 typo: “on study”

·        Line 318 typo: “that CTCs”

Author Response

We appreciate reviewer's comments:

·        Lines 90-92: “CTCs are identified via nucleic acid amplification of mRNA for epithelial cellular adhesion molecule (EpCAM), carcinoembryonic antigen (CEA) or cytokeratin (16), or further immunolabeling of specific surface antigens”  -- consider reworking this sentence for accuracy

·        Line 309 typo: “on study”

·        Line 318 typo: “that CTCs”

All advised  edits have been made in this version of the manuscript.

Reviewer 2 Report

Well revised, acceptable for publication

Author Response

Thanks for the positive response. We appreciate taking time to review our manuscript.

Reviewer 3 Report

All previous concerns have been addressed.

Author Response

(The authors gave the same response as above.)
